# A Regularized Raking Estimator for Small-Area Mapping from Forest Inventory Surveys

**Nicholas N. Nagle [1,\*], Todd A. Schroeder [2] and Brooke Rose [1,†]**

[1]   Department of Geography, University of Tennessee, Knoxville, TN 37996; USA; mrose048@ucr.edu
[2]   USDA Forest Service, Southern Research Station, Knoxville, TN 37910; USA; todd.schroeder@usda.gov
[\*]   Correspondence: nnagle@utk.edu
[†]   Current address: Botany and Plant Sciences, University of California, Riverside, CA 92521, USA.

**Abstract:** In this paper, we propose a new estimator for creating expansion factors for survey plots in the US Forest Service (USFS) Forest Inventory and Analysis program. This estimator was previously used in the GIS literature, where it was called Penalized Maximum Entropy Dasymetric Modeling. We show here that the method is a regularized version of the raking estimator widely used in sample surveys. The regularized raking method differs from other predictive modeling methods for integrating survey and ancillary data, in that it produces a single set of expansion factors that can have a general purpose which can be used to produce small-area estimates and wall-to-wall maps of any plot characteristic. This method also differs from other more widely used survey techniques, such as GREG estimation, in that it is guaranteed to produce positive expansion factors. Here, we extend the previous method to include cross-validation, and provide a comparison to expansion factors between the regularized raking and ridge GREG survey calibration.

**Keywords:** FIA; forest inventory; small-area estimation; survey weight

## 1. Introduction

Surveys for inventories of forests, such as the United States Department of Agriculture Forest Service (USFS) Forest Inventory and Analysis (FIA) program, are typically designed to provide reliable estimates of characteristics over large spatial units, such as states. Managers, however, often desire estimates over much smaller units, or even continuous ("wall-to-wall") maps of natural resources. Unfortunately, the high cost of sampling prevents the ability to create direct survey estimates at such high resolutions.

With the increasing availability of wall-to-wall, remotely-sensed data and the computational power to process these data, there has been a great deal of research toward providing new estimators capable of utilizing both field-based survey data and auxiliary spatial data to produce high-resolution, wall-to-wall maps of forest characteristics. Two types of methods have primarily been used to produce wall-to-wall maps from plot data—these include: (1) empirical predictive models, such as classification and regression trees and Random Forests [1–4], logistic regression [5,6] and probabilistic graphic models [7], which combine satellite-derived composites, GIS layers, and FIA plot data to predict maps of FIA attributes such as forest biomass [8,9], forest type [10], and cause of disturbance [4,11]; and (2) interpolation techniques, such as k-Nearest Neighbors Classification [12,13], gradient nearest neighbor (GNN) [14], and kriging [15], which combine satellite-derived products and FIA plot data to impute maps of FIA attributes, such as forest biomass [16–18], stand density and volume [19], and tree species distribution [20].

Although the two methods differ statistically, the common theme is that they are both highly tailored to produce reliable estimates for a single response variable. While these approaches can

certainly produce reliable results, they can suffer from two possible shortcomings for general use by public agencies. Firstly, models that are tailored to estimating a specific variable may not produce reasonable results when applied to other variables—thus, greater specificity may come at a loss of generality. Secondly, these predictive models often produce estimates that are inconsistent across spatial scales or with "official" published estimates. For example, an empirical model that produces a wall-to-wall map of land use or basal area may not add up to state-level official estimates of land use or basal area. Such inconsistencies can create obstacles for the use of such estimates in official business.

In the eastern United States, the USDA FIA survey uses a permanent network of plots that are visited on a rotating five-year cycle. These plots have been selected through a spatially stratified random design with a density of approximately one plot per 2400 ha (6000 acres). This sampling density certainly does not permit detailed resource mapping, and for most of the United States, does not even permit reliable direct estimates of county averages. For official reporting, the FIA has created custom *survey units*, which are agglomerations of counties that typically contain hundreds of forested sample plots.

To illustrate the approximate scale of these sample densities and survey units, Figure 1 shows a map of the FIA plot locations in South Carolina, as well as county and survey unit boundaries (the locations have been adjusted by the FIA to preserve the confidentiality of landowners). The entire state of South Carolina is divided into only three survey units. While the survey units do roughly correspond to ecological zones, they are much larger than a single county and too large for many desired uses by forest managers.

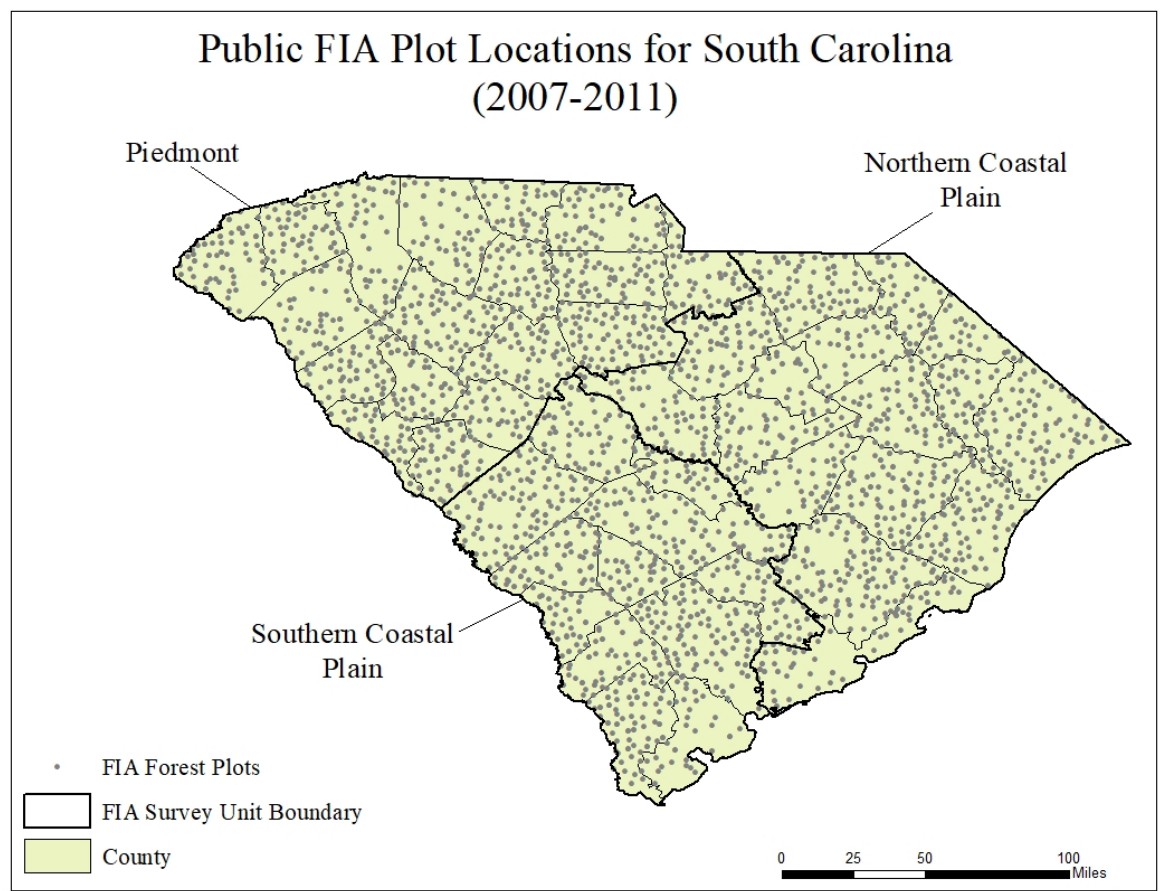

**Figure 1.** FIA plot locations and boundaries of counties and survey units. (Plot locations have been jittered to protect privacy.) South Carolina has three survey units. In 2011, the smallest survey unit in South Carolina contained 672 surveyed plots with some forest use, whereas the smallest county contained only 25 such plots.

In addition to the official publications of characteristics for survey units, the FIA has also published a public version of the plot- and tree-level data (with suitable anonymization). Published along with these plot-level data are survey weights, also called *expansion factors* [21]. These expansion factors allow users inside and outside the USFS to produce custom, large-area estimates that are not otherwise published. By using these expansion factors, users are guaranteed to produce estimates that are consistent with other official estimates. Unfortunately, these expansion factors are designed for estimates at the spatial scale of the survey unit and are not appropriate for use on small areas. Additionally, these expansion factors have been designed to estimate the area of a forest within a survey unit, and may not be suitable for estimates of other FIA variables, such as the hectare of a pine forest in a survey unit or tons of softwood biomass in Colleton County, South Carolina.

In this article, we present a method for producing expansion factors more suitable for creating small-area estimates and wall-to-wall maps of survey characteristics. This method is designed to:

- Accommodate multiple sources of ancillary information;
- Be applicable to all survey characteristics;
- Allow mathematical consistency with published estimates at large-scales;
- Maintain reasonable properties regarding the increase of uncertainty and variance that can be expected with small-area estimation.

Currently, expansion factors only support direct estimation at the scale of FIA survey units (as in Figure 1). In this paper, we define "small area" to be any region smaller than can be supported by a direct estimate from plot-level data. In the context of the FIA, that includes estimates for regions as large as the county; however, we will be producing expansion factors for homogeneous patches that are smaller than counties.

The method we use was originally developed in the geography literature as a modification of existing *dasymetric mapping* techniques used to produce wall-to-wall maps from tables published by the US Census Bureau [22]. Here, we tie that method to existing survey estimation techniques used by FIA, known as raking estimators, and call the new method "regularized raking".

The product of our regularized raking method is a set of small-area expansion factors, $w_{it}$ that match each database record (plot) $i$ to each small area, $t$. Conceptually, these expansion factors also represent a probability distribution of records for each small area—that is, $\frac{w_{it}}{\sum_i w_{it}}$ represents the probability that plot $i$ represents place $t$. Using these expansion factors, it is possible to produce small-area estimates for every characteristic in the FIA database. A wall-to-wall map can be produced by estimating expansion factors for image pixels or other small areas.

The research most similar in objective to ours are the ones by Ohmann and Gregory [14] and Riley et al. [23], who fit GNN and random forests, respectively, to determine a single plot that represents each pixel the most. Regularized Raking, however, rather than providing a *single* best plot for each place, provides a *distribution* across the sample plots for each place to provide information not only about the "best" plot, but how likely each plot is to represent that pixel. Additionally, instead of producing a single expansion factor representing the plot's contribution to the survey unit, our method provides a map of expansion factors for each plot. Such a map would describe the probability that any location can be represented by a particular plot.

## 2. Background

Our proposed method is a modification of the "post-stratification" and "raking estimator" widely used in survey estimation, including the FIA program. We will briefly introduce the raking estimator, but will then describe a different survey weighting approach—Generalized Regression (GREG)—that has been more fertile ground for recent innovation than raking. Some of these innovations to GREG—particularly the ridge [24] and the "least absolute shrinkage and selection operator" (LASSO) [25] regression—served as motivation for our modified raking estimator. After presenting our survey weighting estimator, we will then discuss related work.

*2.1. Survey Estimation*

First, a finite population estimator for the population and study area $U$ of size $N$ is sought, using a sample survey $s$ of size $n$. Let the study area be divided into small, non-overlapping grid cells (such as image pixels) or patches $\{T_t\} \subset U$. Let $\{J_j\}$ be another set of areas, large or small, and possibly overlapping, for which estimates are desired, such as counties, the entire study area, management areas, or the individual pixels. Let $\{y_{ik}\}$ denote the measurement of characteristic $k$ for individual (or plot) $i$. We are interested in obtaining estimates of population totals $\tau_k = \sum_{i \in U} y_{ik}$ and small-area totals $\tau_{jk} = \sum_{i \in J_j} y_{ik}$.

Most survey estimates of population totals can be reduced to linear estimators: $\hat{\tau}_k = \sum_{i \in s} w_i y_{ik}$, where $w_i$ are survey weights or expansion factors [21]. Expansion factors earn their name from the fact that they can be understood as the number of elements in the population that are represented by a sample unit. For example, expansion factors in the FIA data represent the number of acres each plot represents in the entire sample population. For small-area estimation, we will require expansion factors $w_{it}$ that allocate sample $i$ to target patch $t$.

If a survey is conducted with known sampling probability, then a simple, design-unbiased estimator for a population total is the "Horvitz–Thompson" (HT) estimator [26,27]:

$$\hat{\tau}_j^{HT} = \sum_{i \in s} d_i y_{ij} = \mathbf{d}' \mathbf{y}_j$$

where the weights are called *design* weights because they are related to the sample design through the inverse sampling probability, that is, $d_i = 1/\pi_i$. In a simple random sample with equal probability, the expansion factors are simply $\frac{N}{n}$.

At the next level of complexity, survey weights are often modified to account for known ancillary data. For example, the FIA program adjusts the HT weights to account for forest, non-forest, and forest-fringe land areas, as derived from the National Land Cover Database [28–30].

2.1.1. The Raking Estimator

Two widely used methods to account for ancillary data are *raking* (also called "iterative proportional fitting", or "post-stratification" if there is only one ancillary variable) and GREG [31]. Both methods modify (or calibrate) the HT expansion factors so that the survey estimates reproduce known population totals, $\tau_\ell = \sum_{i \in U} x_{i\ell}$. (We used '$\ell$' and '$x_{i\ell}$' to indicate auxiliary variables, and '$k$' and '$y_{ik}$' to indicate other plot-level characteristics that we would like to estimate or map.) In particular, both methods create weights such that $\sum_{i \in s} w_i x_{i\ell} = \tau_\ell$ for each ancillary total. Thus, both raking and GREG assume that ancillary totals are precisely known.

When we say that these methods assume that an auxiliary variable is measured without error, this means that an auxiliary total is identical to what would be obtained if the sample survey was a 100 percent sample instead, that is, $\tau_\ell = \sum_{i \in U} x_{il}$. If this is not the case, then raking and GREG are not consistent estimators. Auxiliary data, however, almost always contain uncertainty relative to the survey variable. For example, ancillary estimates of land cover or tree canopy cover are statistical predictions developed from measurements of reflectance at the satellite sensor. Additionally, field surveys contain their own sources of measurement and other non-sampling errors which will cause misalignment with auxiliary data. Hence, uncertainty in ancillary data seems to be the standard, not the exception.

Raking is an iterative process that uses the HT estimates to initialize the expansion factor for each survey plot. Afterwards, an estimated ancillary total $\sum_i w_i x_{i\ell} = \hat{\tau}_\ell$ is calculated, and then the expansion factors are all adjusted by the multiplicative factor $\tau_\ell / \hat{\tau}_\ell$. This guarantees that the known ancillary total $\tau_\ell$ is accurately reproduced by expansion factors. This step is then repeated for each of the ancillary totals, and the entire process is repeated from the beginning until the expansion factors converge, or until the algorithm reaches a maximum number of iterations. The raking method is used by FIA;

however, it converges in one step since there is only one ancillary variable—a spatial layer of forest, non-forest, and forest fringe area. This layer is derived from NLCD land cover, NLCD Tree Canopy Cover, and land ownership.

A naive method to generate small area estimates would be to simply incorporate many ancillary totals for small areas, $\{T_t\}$. In practice, however, this produces erratic expansion factors and may cause the raking algorithm to fail to converge. Our regularized raking method is a natural extension of the raking estimate that does not succumb to these problems as easily. While our estimator is an extension of the raking estimator, it is motivated by recent innovations to the GREG estimator—thus, next we will be discussing GREG.

### 2.1.2. The Generalized Regression Estimator

The GREG estimator was motivated by a linear regression problem using the unknown total $\tau_k$ as the response variable, and the variables $\tau_\ell - \tau_\ell^{HT}$ as the explanatory variables. Intuitively, if the HT estimator over- or under-predicts some of the ancillary totals, then the weights may need to be adjusted accordingly. The GREG estimator is given by the equation:

$$\hat{\tau}_j^{GREG} = \hat{\tau}_j^{HT} + \sum_\ell (\tau_\ell - \hat{\tau}_\ell^{HT})\beta_\ell$$

where the $\beta_\ell$ is the weighted least squares regression coefficient obtained by regressing the response variable $y_j$ on the ancillary variables $(\tau_\ell - \hat{\tau}_\ell^{HT})$, using the design weights $d_i$ on each observation.

It may appear that the GREG estimator depends on the response variable $y_j$, and thus that a different estimator would be needed for each response variable, but it is possible to rewrite the GREG estimator in terms of an expansion factor as:

$$\hat{\tau}_j^{GREG} = (\mathbf{w}^{GREG})'\mathbf{y}_j = (\mathbf{d} + \boldsymbol{\lambda}'(\boldsymbol{\tau}_x - \hat{\boldsymbol{\tau}}_x^{HT}))'\mathbf{y}_j, \tag{1}$$

where $\boldsymbol{\lambda}$ are coefficients that do not depend on the response $y$. Thus, the expansion factors $\mathbf{w}^{GREG}$ can be calculated once and stored for general use.

The GREG estimator is not widely used in survey practice. In particular, the GREG estimator has a tendency to produce some negative expansion factors. This is especially common when many ancillary data are used. In these same situations, the raking estimator produces non-negative but sometimes erratic expansion factors, whereas the GREG estimator tends to produce expansion factors that are slightly less erratic, but sometimes negative. Negative survey weights are clearly problematic, as they defy the interpretation of the weights as expansion factors. For this reason, GREG is rarely used in situations in which agencies must publish the expansion factors.

### 2.1.3. Regularized GREG

A recent expansion of GREG that motivated our estimator was that of regularized regression, including ridge and LASSO regression. As mentioned earlier, GREG and raking can often produce erratic results when there are many ancillary totals. Ridge regression modifies the GREG estimator by adding a term $\sum_\ell \beta_\ell^2$ to the least squares regression problem [24]. This has the effect of inaccurately reproducing ancillary totals if doing so causes the coefficient to become larger. LASSO regression modifies the GREG estimator by the addition of the term $\sum_\ell |\beta_\ell|$ to the least squares regression [25]. Ridge and Lasso are both types of "regularization" estimators. Regularization has proven to be a powerful heuristic tool for prediction problems involving many explanatory variables because it effectively addresses common practical problems caused by the multicollinearity and overfitting problems that emerge in such situations. Regularization estimators typically trade off a little bias in a prediction in exchange for greatly reduced mean square error.

The survey not being required to accurately reproduce the ancillary data is an attractive property, both conceptually and statistically. Conceptually, we ought to recognize that ancillary data do not

usually represent a "gold standard" of truth for the population total. For example, the FIA weighting process uses estimates of the forest, not-forest, and forest edge that are derived from satellite imagery. Satellites do not observe forests—they observe spectral reflectances, which we then feed through models that create predictions of tree canopy and land cover. Thus, while we may have confidence that the model is, on the whole, reliable, the output by no means represents the absolute truth. By forcing our survey estimates to match these modeled outputs, we are forcing our survey estimates to reproduce the errors found in these imperfect spatial data layers.

We believe that many of the practical limitations of calibration estimators are amplified by calibrating on noisy and imprecise auxiliary variables. Intuitively, if the auxiliary data contain an element of truth (or signal) and an element of error (or noise), then the calibration will incorrectly reproduce the noise along with the signal. This places limits on the number of auxiliary variables that can be used—if too many noisy auxiliary variables are used, then calibration will overfit these data and the final weights will be erratic. In regression modeling, the effects of overfitting are similar to the effects of multicollinearity.

To extend these ideas to raking, we first point to the fact that GREG solves the optimization problem [32]:

$$\min_{w_i} \sum_{i \in s} \frac{1}{2} \frac{(w_i - d_i)^2}{d_i} \text{subject to} \sum_{i \in s} w_i x_{i\ell} = \tau_\ell .$$

Thus, GREG can be interpreted as "find new expansion factors $w_i$, that are close to the HT factors but that also satisfy the ancillary constraints".

Guggemos et al. [24] showed that the ridge GREG estimator can be similarly written as:

$$\min_{w_i} \sum_{i \in s} \frac{1}{2} \frac{(w_i - d_i)^2}{d_i} + \frac{1}{2\gamma} \sum_\ell c_\ell (\tau_\ell - \sum_i w_i x_{i\ell})^2 .$$

The regularization parameter $\gamma$ serves to regulate the tradeoff between finding expansion factors that are close to the unbiased design weights and finding expansion factors that reproduce the auxiliary totals. In contrast to GREG, which accurately reproduces the ancillary totals, the ridge GREG estimator only approximately reproduces the ancillary totals. The approximation can be made exact by increasing $c_\ell$ or decreasing $\gamma$. Whereas $\gamma$ is a global regularization factor, the factors $c_\ell$ provide a means to control the relative importance of reproducing specific ancillary totals.

### 2.1.4. Regularized Raking

To develop our regularized raking estimator, we first point to results showing that the raking and GREG estimators are similar. Deville and Sarndal [33] proposed a class of estimators that calibrate the Horvitz–Thompson design weights to ancillary data. These estimators are:

$$\min_{w_i} \sum_{i \in s} d_i G(w_i, d_i) \text{ subject to } \sum_{i \in s} w_i x_{i\ell} = \tau_\ell$$

where $G(w_i, d_i)$ is a generalized distance measure. They show that when the distance function is the chi-square function $(\frac{w_i - d_i}{d_i})^2$, then the GREG estimator results. Similarly, when the distance function is the entropy distance function $w_i \log(\frac{w_i}{d_i})$, then the raking estimator results. In both cases, minimizing the distance between the HT weight and the calibrated weights preserves an approximate design-unbiased property [24]. For our purposes, an advantage of the entropy distance function compared to the chi-square function is that it prevents survey weights from becoming negative.

By analogy to the regularized GREG estimator, we propose a regularized raking (or regularized entropy) estimator. Let $\tau_{j\ell}$ be an ancillary total for spatial region $J_j$ and characteristic $k$. The estimator we use is

$$\min_{w_{i\ell}} -\sum_{i \in s} \sum_t w_{it} \log(\frac{w_{it}}{d_{it}}) - \frac{1}{2\gamma} \sum_\ell \sum_j \frac{(\tau_{j\ell} - \sum_{i \in s} \sum_{t \in J_j} w_{it} x_{i\ell})^2}{\sigma_{j\ell}} .$$

The output of this estimator is the set of strictly positive expansion factors $w_{it}$. Note that the single plot-level expansion factor $w_i$ has been expanded to a vector of expansion factors, $w_{it}$. These expansion factors have units of acres of patch $t$ represented by plot $i$. If the weights are normalized by the small-area's size $\frac{w_{it}}{\sum_t w_{it}}$, then they create a sort of empirical histogram or probability density across the samples for each small patch.

Nagle et al. [22] suggested setting $\sigma_{j\ell}$ to the standard deviation of the ancillary total $\tau_{t\ell}$, as this serves to distribute the accuracy in proportion to the quality of the ancillary data. The estimator will try to accurately calibrate the weights to the ancillary, but if there are conflicts between the ancillary data, then the weights will lean toward a closer reproduction of the more precise ancillary data.

Nagle et al. also suggested setting the prior weights $d_{it}$ to $d_i \left( \frac{\text{area of } T_t}{\text{area of } U} \right)$. This has the effect of assuming that, in the absence of any ancillary information, the best prior estimate is that each patch is identical. Other prior weights may also be possible, such as by weighting nearby samples more, but this is not explored in this article.

The regularization factor $\gamma$ was not considered by Nagle et al. We include it here to allow a data-driven selection to the regularization. If $\gamma = 0$, then the final weights will be usage of the HT weights at each patch. If $\gamma$ is large, then the final expansion factors will try to reproduce the ancillary totals even if it requires large deviations from the HT weights. Later, we describe a cross-validation procedure to choose this regularization factor.

## 2.2. Related Work

Regularization is an approach that is experiencing growing exploration in survey research. In addition to ridge regularization, which uses a square error penalty as in this paper, recent research has explored LASSO regularization, which uses an absolute deviation penalty [25,34]. McConville et al. reported that LASSO estimators, while producing fewer negative expansion factors than GREG, still sometimes produce negative factors. This problem renders them inadequate for creating general-purpose expansion factors.

More generally, it may be possible to use raking with a LASSO regularization rather than a ridge regularization. LASSO is often justified over ridge regression on the grounds that it is "sparse" and automatically selects some ancillary variables which can be used and others which cannot be used. Sparsity is a useful feature when the regression coefficients $\beta$ must be stored for later use. However, we find the justification less compelling in the current case because the model only needs to be fit once, and it is not a set of regression coefficients that must be stored, but expansion factors $w$. LASSO has no effect on the sparsity of expansion factors $w$, and here there is no guarantee that a sparse set of regression coefficients will lead to more efficient or robust expansion factors. A sparse set of expansion factors may be desirable, but neither ridge nor LASSO methods currently provide that.

Another similar approach is that which was previously mentioned by Riley et al. [23]. Using ancillary data on vegetation and biophysical attributes, they built a random forest model to identify the single FIA plot that best predicts the ancillary data at each 30 m pixel across the Western US. This is in contrast to our expansion factor approach that identifies the probability that each site is represented by each plot. Additionally, Riley et al. fit their model at a large-scale, and their model often identified the best plot which was thousands of kilometers away from the pixel. In our implementation, we kept the plots limited to the most immediate FIA survey unit; however, it is theoretically possible to expand our approach to a larger region, as did Riley et al.

A similar product is the National Tree-List Layer [35]. That product creates a geographic layer of trees for each site by stratifying places using vegetation and biophysical characteristics, and then a nearest neighbor search from the site to a database of sample plots that includes the FIA and other samples.

## 3. Data

### 3.1. Forest Inventory and Analysis

The sampling network of the FIA is a non-overlapping mesh of 2400 ha ($\sim$5930 acres) hexagons [21]. A permanent sample plot is randomly located within each cell, which is visited on a 5- to 7-year cycle in the Eastern US. Our study region is the state of South Carolina, which is divided into three survey units (refer back to Figure 1). Following FIA practice, we will estimate expansion factors separately for each survey unit.

When a plot is sampled, the site is first viewed in the office using aerial imagery. If the office staff determine that the site is unlikely to be in use as a forest, then its land use is recorded and the sampling process ends for that site. If the plot is regarded to possibly contain use as a forest, then the plot is visited by a field crew. The field crew divides each plot into separate "condition classes" so that each condition represents a homogeneous land use type and ownership class. In this study, we were interested in three attributes: land use, forest basal area, and forest volume. Land use classification is not nationally standardized, and so we used the classification method used by the Southern Research Station (coded as LAND_USE_SRS in Burrill et al. [30]). We broke down the land use classification into the following categories: forest, other agriculture, urban, and other uses.

FIA field crews also recorded the dominant species type of forests found in each condition class (FORTYP). The Forest Type is a three-digit classification which we collapsed into a two-digit classification corresponding to "Pine and other Softwood", Oak/Hickory, Oak/Pine, Oak/Gum/Cypress, Elm/Cottonwood, and Other. Owing to the small sample sizes, we further combined the Oak/Gum/Cypress and Elm/Cottonwood into one group, as well as the "Other" category with the Oak/Hickory. This produced a classification of four different species groups.

We then combined the Land Use and Forest Type variables into one nested variable, so that Forest Use was divided by Forest Type, and non-forest Use was divided by the Use class. The definitions of these nine classes are shown in Table 1.

**Table 1.** Table of notation.

| Notation | Description |
|---|---|
| $i$ | index of (possible) location |
| $s$ | the set of survey locations |
| $n$ | the number of survey locations |
| $U$ | Study domain (the set of locations in the population) |
| $N$ | the size of the study domain $U$ |
| $t$ | index of patches |
| $T_t$ | Patch $t$ (a spatial unit) |
| $j$ | index of auxiliary total |
| $\tau_j$ | auxiliary total |
| $\sigma_j$ | standard deviation of $\tau_j$ |
| $J_j$ | Auxiliary region $j$ (a spatial unit) |
| $k$ | index of survey characteristic |
| $y_{ik}$ | survey measurement of characteristic $k$ at location $i$ |
| $\ell$ | index of auxiliary characteristic |
| $y_{i\ell}$ | survey measurement of characteristic $\ell$ at location $i$ |
| $d_i$, $d_{it}$ | design weight |
| $w_i$, $w_{it}$ | expansion factor |
| $\gamma$ | regularization factor |

We also estimated the Forest Volume (VOLCFNET) for each plot, which is defined as the (estimated) net volume of wood in timber species greater than (or equal to) 5 inches in diameter at breast height (DBH).

In addition to land use, forest type, and forest volume, we also used the estimate of Basal Area of Live trees (BALIVE) for each plot for prediction. We did not use basal area as an ancillary variable,

but as a response variable for creating small-area estimates. BALIVE represents the density of trees greater than or equal to 1 inch DBH sampled in each condition (reported in units of square feet per acre). For creating survey estimates, we set BALIVE to 0 in sites that were non-forested or that did not contain any trees larger than 5 in.

In addition to the FIA variables, we used land cover and tree-canopy cover maps from the 2011 National Land Cover Database (NLCD2011) to generate our ancillary totals [36].

## 4. Methods

To implement the weighting strategy, we required auxiliary estimates of quantities that could be estimated for the FIA plots. For example, it was possible to obtain the NLCD land cover class for each plot in the office. However, to demonstrate the flexibility of the model, we instead used the NLCD land cover and tree-canopy cover layers in Bayesian generalized additive models to develop wall-to-wall maps of: (1) the nine-class land use and forest type classification in Table 2, (2) two-class forest/non-forest use, and (3) forest volume. For the purpose of this article, the relevant factor is that these models generate predictions and standard errors for forest characteristics measured at FIA plots. While we created small-area estimates for basal area in addition to those above, we have not included a direct measurement of basal area as an ancillary layer.

Theoretically, we could have estimated expansion factors for individual pixels; however, this is computationally infeasible and unnecessary. Expansion factors for pixels that are observationally equivalent (i.e., that have the same predictions and standard errors) will also be identical. Thus, it not necessary to fit the model at the pixel scale, but at a scale that has the pixels grouped into patches that are homogeneous with respect to the ancillary predictions and standard errors. Since our ancillary maps were derived from NLCD land cover and tree-canopy cover layers, and we also desire county-level estimates, we grouped pixels into patches based on county, NLCD class, and tree-canopy cover (grouped into 20 bins as 0–4 percent, 5–9 percent, ..., 95–100 percent). For each of the three survey units, there were 180 such patches (nine land-cover classes multiplied by 20 tree-canopy cover classes). When these homogeneous patches were overlayed on the 46 counties of South Carolina and empty combinations were deleted, there were 8260 such patches across the state of South Carolina. These are the target patches $T_t$ defined in the background section.

**Table 2.** Custom Land Use/Forest Type classification. Variable names refer to those in the FIA Database [30].

| Group | Class | FIA Database Definition |
|---|---|---|
| Forest | Eastern Softwood | LAND_USE_SRS in (1,2) AND FORTYPCD in (100–199) |
| | Oak/Pine | LAND_USE_SRS in (1,2) AND FORTYPCD in (400–499) |
| | Oak/Hickory | LAND_USE_SRS in (1,2) AND FORTYPCD in (500–599; 800–998) |
| | Bottomland Hardwood | LAND_USE_SRS in (1,2) AND FORTYPCD in (600–699) |
| Not Forest | Not Stocked | LAND_USE_SRS in (1,2) AND FORTYPCD in 999 |
| | Agriculture | LAND_USE_SRS in (10–19) |
| | Urban/Developed | LAND_USE_SRS in (30–39) |
| | Barren | LAND_USE_SRS in (40–49) |
| | Water/Wetland | LAND_USE_SRS in (90–99) |

The mapped outputs from these models were then used as auxiliary variables in the regularized raking estimator. In all cases, the ancillary data model produces both estimates, as well as standard errors of prediction, which are used to weight the ancillary variables.

Figure 2 shows an example model fit for the auxiliary variables: the probability of forest use and the probability of the Eastern Softwood Forest Type, as well as their standard errors, for one of the survey units. It is obvious that the standard error of the "Forest" land use class is smaller than the standard error for the more specific "Eastern Softwood" class. This is especially so at the higher canopy cover levels that are most likely to be forested.

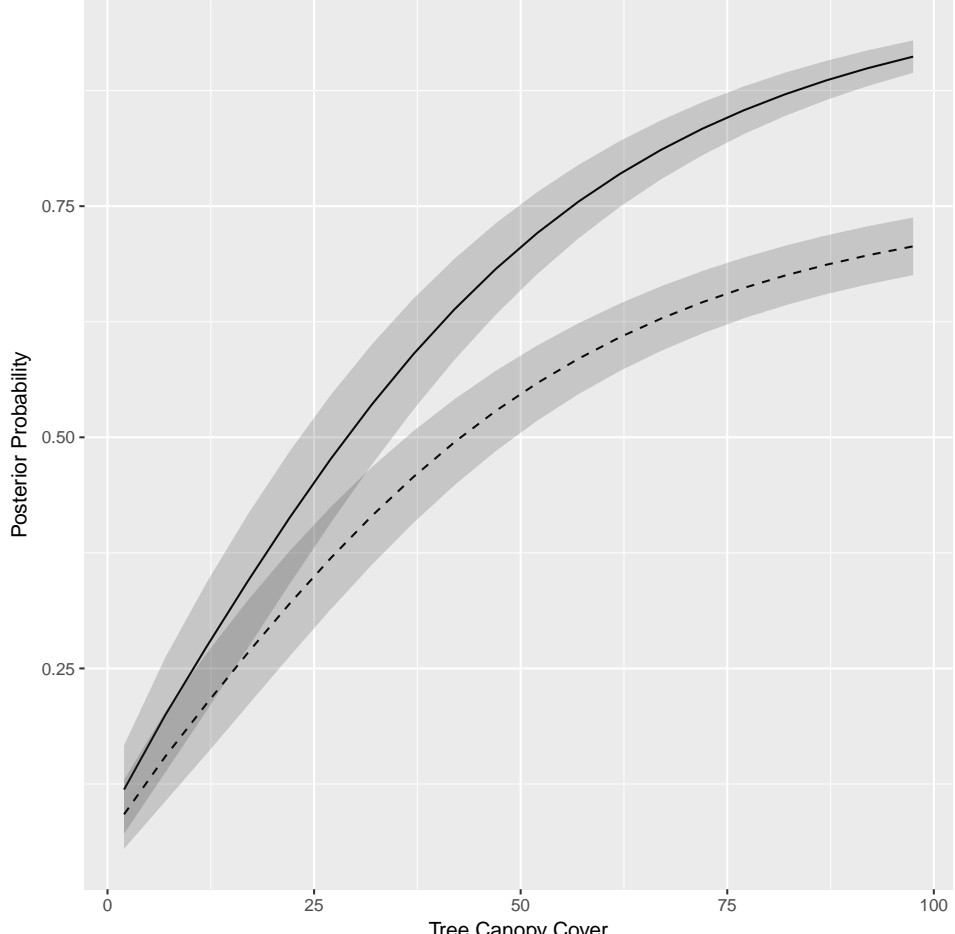

**Figure 2.** Posterior probabilities and standard error for Forest (solid) and Eastern Softwood (dashed). Only NLCD class 42 in the Piedmont survey unit is shown. While auxiliary data were predicted or imputed from a model such as that used here, it is obvious that there is some uncertainty in the prediction.

The complete set of auxiliary variables used across the state include:

1.  Patch level: estimates of the nine-class land use/forest-type classification (180 unique patches $\times$ 9 classes $\times$ 3 survey units = 4860 ancillary totals);
2.  Patch level estimates of the two-class forest/non-forest land use (180 $\times$ 2 $\times$ 3) = 1080 ancillary totals);
3.  Patch level estimates of forest volume (180 $\times$ 3 = 540 ancillary totals);
4.  County size (45 ancillary totals), where the variance of county size was set to 0 so that the expansion factors could reproduce them accurately.

It may appear that there are some redundancies in the set of auxiliary variables. For example, the land areas of the nine-class map perfectly recreates the land areas of the two-class map, and the number of acres in the two classes must add up to the number of acres in the county. When the uncertainties of the auxiliary data are considered during regularization, however, they are no longer redundant. For example, while there is uncertainty in the number of acres of forest and number of acres of non-forest land, there is zero uncertainty in their sum across the county. Also where there is relatively large uncertainty in each of the individual nine-class land-use/forest-type proportions at each patch, there is less uncertainty in their aggregation to the two-class forest/non-forest proportions (e.g., see Figure 2). The regularization incorporates ancillary data variance, but not covariance. It is possible that the nine-class map would be sufficient if all cross-covariances are accounted for, but this would

dramatically complicate the computational routines. Additionally, while it is somewhat reasonable to obtain estimates of prediction error variance from ancillary geospatial data products, it is much less reasonable to expect covariances. For example, while we might obtain published standard errors for pixel-level estimates of tree-canopy cover, we will not get covariances between pixels, nor will we get covariances between tree-canopy estimates and land cover estimates (for example). Of course, many such ancillary predictions will be correlated, but we must disregard that aspect here.

Using the ancillary maps created by our predictive model, we fit both a regularized raking model and a ridge GREG model to each of the three FIA survey units. The regularized raking model was fit using a custom R package available from the first author. The ridge GREG model was fit using the 'Matrix' package in R using the equation given in ([24] p. 3204).

For both the raking and GREG estimators, the regularization parameter $\gamma$ was determined using cross-validation. To conduct the cross-validation experiment, we first divided the survey plots into a training dataset and a test dataset by withholding 1/16 of the plots from training. These withheld plots were a subset of the FIA plots located on a coarser hexagonal grid, and thus represent a spatially systematic test set. To mimic the uncertainty in the auxiliary data, we simulated $N = 30$ instances of the auxiliary totals. Thirty sets of expansion factors were then estimated using the training data and each of the auxiliary sets, and these expansion factors were then used to predict forest volume at the test locations. The cross-validation error was obtained by averaging the prediction error across the test locations and the simulations. This was repeated across a range of the regularization parameter $\gamma$, and the value with the lowest square error was selected.

Before continuing, we remark that our ancillary data were fit using a model that used the survey estimates as inputs. Thus, in our example, the auxiliary data are endogenous to the sample. Breidt and Opsomer [37] considered endogenous post-stratification and suggested that this had little effect on the final survey estimates. How that result translates to the small-area situation we consider here, however, is unknown.

## 5. Results

### 5.1. Comparison of Raking and GREG Expansion Factors

A benefit of regularized raking is that it produces positive expansion factors, while ameliorating erratic weights that are caused by overfitting to inconsistent and noisy ancillary data. Table 3 shows the minimum and maximum expansion factors for the ridge GREG and regularized raking estimators as a function of the regularization parameter $\gamma$. Recall that, as $\gamma$ approaches zero, the normal GREG and raking estimators are obtained. In contrast, as $\gamma$ gets large, both estimators converge on the HT expansion factors. The minimum expansion factor for the ridge GREG model is consistently negative for all but the largest values of the regularization factor $\gamma$. In contrast, the raking estimator always produces strictly positive expansion factors.

It is harder to interpret the absolute magnitude of the maximum expansion factor because they have units of acres and depend on the size of the patches, but it is easy to compare the maximum weights between the raking and GREG models. The maximum expansion factor is always smaller for the GREG estimator than for the raking estimator, and the largest expansion factor can get quite large for very small values of the regularization parameter (that is, when we try to precisely match the auxiliary variables). When the regularization parameter is very small, the regularized raking estimator converges to the raking estimator, and we note that the normal raking estimator is not guaranteed to exist when there are incompatible auxiliary variables. Our simulations revealed some explosive behavior in the regularized raking estimator for very small values of gamma ($\gamma < 0.01$). Regularization appears to temper these undesirable effects.

Figure 3 shows the cross-validation error as a function of the regularization parameter $\gamma$. The cross-validation measure is the average square error of estimating forest volume across the withheld plots. Regularized raking consistently performs better than ridge GREG when predicting

forest volume out-of-sample. This is especially interesting since it is GREG that minimizes (within-sample) square error loss, not raking. The improvement seems to be in the fact that GREG can produce errors beyond the range of the data, whereas raking cannot. We also note that regularized raking prefers slightly more regularization than does ridge GREG. This is slightly unfortunate—since GREG is less computationally demanding, and cross-validation is a computationally intensive process, we had hoped that the optimal regularization parameter for GREG would also be a good regularization parameter for raking. One possible explanation for why raking may prefer larger regularization parameters stems from the more extreme maximum weight of the raking procedure relative to GREG.

**Table 3.** Minimum and maximum expansion factors for the ridge GREG models and regularized raking. South Carolina, Survey Unit 1.

| Gamma | GREG Weights | | Raking Weights | |
|---|---|---|---|---|
| | **min** | **max** | **min** | **max** |
| 0.1 | −151.58 | 2381.00 | $<1.00 \times 10^{-8}$ | 2930.64 |
| 0.2 | −113.37 | 2029.45 | $<1.00 \times 10^{-8}$ | 2833.97 |
| 0.3 | −87.17 | 1776.99 | $<1.00 \times 10^{-8}$ | 2744.86 |
| 0.4 | −75.52 | 1586.54 | $<1.00 \times 10^{-8}$ | 2659.87 |
| 0.6 | −76.66 | 1317.79 | $<1.00 \times 10^{-8}$ | 2498.54 |
| 0.8 | −77.93 | 1136.75 | $<1.00 \times 10^{-8}$ | 2346.58 |
| 1.0 | −78.73 | 1006.18 | $<1.00 \times 10^{-8}$ | 2203.08 |
| 2.0 | −78.66 | 672.69 | $4.00 \times 10^{-7}$ | 1604.76 |
| 8.0 | −72.25 | 323.54 | $1.49 \times 10^{-5}$ | 465.47 |
| 30.0 | −49.23 | 203.38 | $4.51 \times 10^{-5}$ | 219.71 |
| 100.0 | $5.75 \times 10^{-5}$ | 161.34 | $5.80 \times 10^{-5}$ | 161.60 |

It should not be inferred from Figure 3 that regularized raking is always better than ridge GREG. At some point to the left of the range in these plots ($\gamma < 0.01$), the regularized raking error explodes as the coefficients fly apart to infinities trying to reach a raking solution that may not exist, whereas the ridge GREG line converges to the normal GREG solution. Furthermore, Figure 3 is slightly misleading as the y-axis does not extend to zero. In terms of root mean square error, the difference between regularized raking and regularized GREG is not as extreme as might appear here. The most noticeable difference to users is likely to be the presence or absence of negative expansion factors and not any differences in estimates.

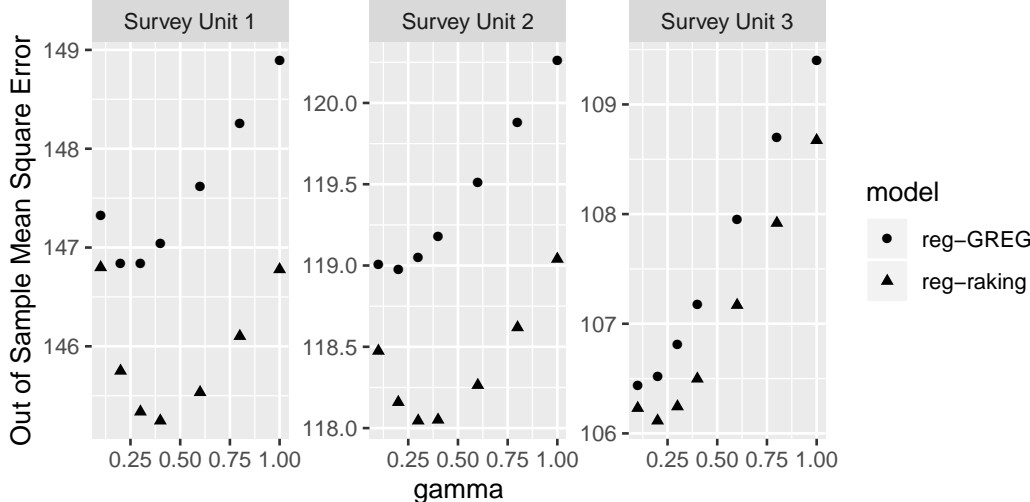

**Figure 3.** Cross-validation error as a function of the regularization parameter $\gamma$ for regularized raking (triangle) and ridge GREG (circle), for survey units 1 (**left**), 2, and 3 (**right**). The response variable is the mean square error for predicting forest volume at out-of-sample locations.

### 5.2. Prediction Maps

Once expansion factors are calculated, they may be used to estimate wall-to-wall maps of any plot characteristic. Figure 4 shows maps of (a) Basal Area for all trees and (b) Basal Area for Eastern Softwood species. At this scale, it can be seen that the maps reproduce large-scale patterns in forest areas, such as the dominance of softwood species in the Piedmont region and relative absence of softwood species from coastal bottomlands.

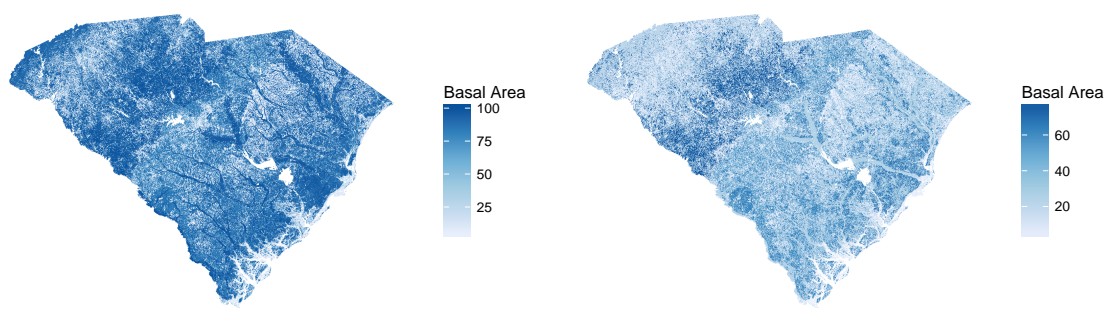

(**a**) All forest types.　　　　　　　　　　　　　　　(**b**) Eastern Softwood forest type.

**Figure 4.** Estimated basal area (sq ft per acre).

Figure 5 shows a higher resolution comparison of the ancillary layers and the raking estimates for a 30 km × 10 km swath in the South Carolina Piedmont. While the regularized raking estimate does not accurately reproduce the ancillary layers, the differences are visually indistinguishable here. It is worth pointing out that the any errors in the ancillary data will be carried through to the raking estimates. For example, the model that creates Forest Volume from NLCD tree canopy cover and NLCD land use saturates with respect to tree canopy cover; while the model is able to distinguish between low and moderate volume, the auxiliary model is unable to effectively model high volume. The range of predictions in the auxiliary model (0 to 100 sq ft per acre) is restricted relative to values of volume on the landscape (0 to 1000 sq ft per acre), and this restricted range in the auxiliary data is duplicated by restricted range in the regularized raking estimates.

While we were able to produce high-resolution wall-to-wall maps, perhaps the best use of the expansion factors may be to produce estimates for slightly larger areas. Table 4 shows the correlation between the direct survey estimates and the regularized raking estimates. At the plot level, the direct estimate is just the observed value. At the county level, the direct estimate is the weighted value of all plots within the county.

As expected, the correlation between the direct estimate and the regularized raking estimate increases as the spatial unit gets larger. Whereas the correlation at the plot level is relatively weak, the correlation increases when estimating larger units, such as counties. We would not expect the county level correlation to necessarily equal 1, because the direct estimates have high sampling variance at the county level, but the results nonetheless suggest that estimates for larger units are more reliable than for the smallest areas.

The distinguishing characteristic of this method is that all estimates use the same expansion factors $w_{it}$. Typically, small-area estimators will fit separate models for each variable of interest. For example, the model for Basal Area might be different from the model for Basal Area of Pine or from the model of Volume. An advantage of using the same weights is that all maps are consistent, and it is impossible to obtain more Basal Areas of softwood species than Basal Areas of all species, and it is impossible to obtain estimates that have a positive Basal Area but no volume, and vice versa. Additionally, all estimates are constrained to the range of the data. Because none of the weights are

negative and they form a convex combination, all estimates lie strictly within the range of the survey data. In particular, negative estimates are impossible, unlike other methods, such as kriging and GREG.

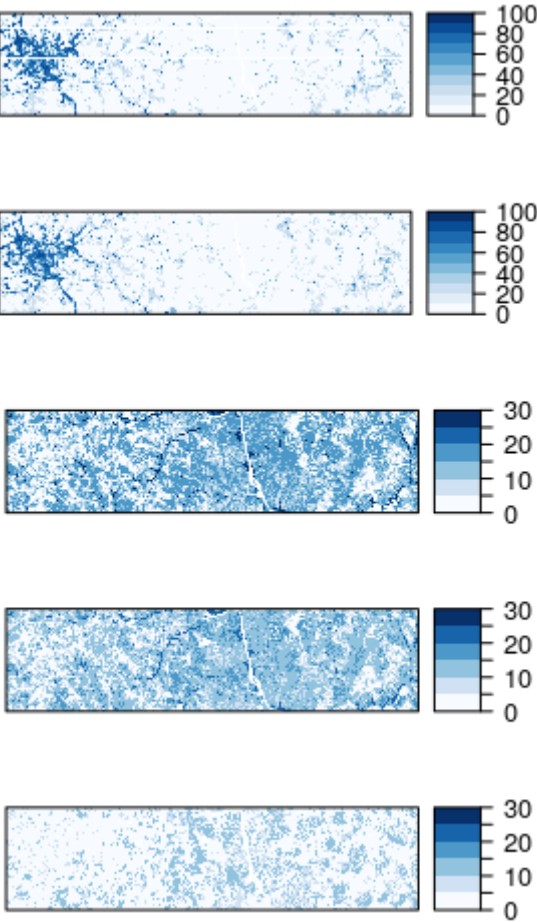

**Figure 5.** Ancillary and raking-estimated maps. Map size is 30 km × 10 km. From top to bottom: Probability of developed land use—ancillary; probability of developed land use—raking estimate; Forest Volume—ancillary; Forest Volume—raking estimate; Volume of Eastern Softwood—raking estimate.

**Table 4.** Pearson correlation coefficient between the direct survey estimate and the regularized raking estimate. At the plot level, the direct survey estimate is the plot measurement (when multiple conditions are present, the direct estimate is the weighted average across conditions). At the county level, the direct survey estimate is the Horvitz–Thompson estimate from the plots in that county.

| Variable | Plot | County |
|---|---|---|
| Forest Land Use | 0.65 | 0.83 |
| Oak Pine Forest Type | 0.49 | 0.86 |
| Eastern Softwood Forest Type | 0.59 | 0.91 |
| Volume | 0.27 | 0.79 |
| Basal Area | 0.68 | 0.88 |

## 6. Conclusions

In this paper, we introduced a new method for producing survey expansion factors that are suitable for small-area use. This method extends post-stratification and raking methods already in use by the FIA and other survey agencies. Raking estimators calibrate the survey probability weights to reproduce ancillary data. The regularization method modifies raking to allow for the approximate

reproduction of ancillary data, with a relative weighting between different ancillary data based on their relative precision. Whereas the usual raking estimator often gets overwhelmed by relatively few ancillary variables, producing erratic and very large expansion factors, the regularized raking estimator in this paper utilized tens of thousands of small-area ancillary totals.

Like many other methods in the literature, the regularized raking method allows the integration of FIA plot data with wall-to-wall ancillary data. Unlike those other methods, however, the regularized raking methods is relatively agnostic about the response variable. It is mathematically possible to use the expansion factors to produce wall-to-wall maps for any characteristics available for FIA plots. These wall-to-wall maps and small-area estimates can also be made to be consistent with published totals (it is even possible to create the published totals from the same expansion factors).

However, the regularized raking method is by no means optimal for any characteristic. A regularized raking estimator for any characteristic is certain to be outperformed by a highly tailored predictive model. For example, even if an "optimal" data layer is used as an ancillary variable, the regularized raking estimator would not exactly reproduce this estimate unless the prediction standard error were artificially forced to be zero, which could lead to erratic expansion factors from the raking estimator.

The regularized raking estimator can have an intriguing scale-dependence, in that it can transition from a direct survey estimate at the scale of the survey unit, similar to the current practice of producing expansion factors, to an "indirect" or model-based estimate at small areas, borrowing samples and statistical relations from across the entire study area to make small-area estimates. This also suggests that the estimates are approximately design-unbiased at the scale of the survey unit, but are less so for small-area estimation. If the plot-level weights are accurately constrained to add to the survey unit-level Horvitz–Thompson weights, then the same set of expansion factors can produce accurate design-unbiased estimates at the survey unit level, and model-based estimates at the small-area level.

Finally, we would like to comment on the feasibility of disseminating these expansion factors. Publishing the expansion factors for one survey unit in South Carolina would require a table containing one row for each of the approximately 3000 plots, and one column for each of the approximately 2700 patches in the survey unit. If the expansion factors are stored at integer precision, such a table would require about 50 MB of storage, which is easily within the range of feasibility for a geospatial dataset. Storage sizes would be much smaller in a suitable binary format (such as in a SQL database table).

Further research is needed to determine a suitable set of ancillary data. These ancillary data should be correlated with as many plot-level characteristics as possible. A benefit of the regularized approach is that many ancillary variables may be considered without experiencing too many undesirable effects from overfitting. Regularization is also an effective strategy for automatic variable selection [34]. Efforts are currently underway to explore the use of ancillary data developed from Landsat time-series-fitting algorithms (e.g., [38–40]). Potentially, combining these smoothed outputs with regularized raking will allow the construction of temporally consistent time-series expansion factors, which can lead to improved estimates of forest characteristics across space and time.

**Author Contributions:** Conceptualization, N.N.N. and T.A.S.; methodology, N.N.N.; software, N.N.N.; validation, N.N.N.; formal analysis, N.N.N. and T.A.S.; data curation, B.R.; writing–original draft preparation, N.N.N.; writing–review and editing, T.A.S. and B.R.; visualization, N.N.N. and B.R.; supervision, N.N.N. and T.A.S.; project administration, T.A.S.; funding acquisition, T.A.S.

**Funding:** Funding for this research was provided by the U.S. Forest Service, Southern Research Station, Forest Inventory and Analysis (FIA) program (via agreement 17-CR-11330145-057).

**Conflicts of Interest:** The authors declare no conflict of interest.

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
