# Peer review of "A Regularized Raking Estimator for Small-Area Mapping from Forest Inventory Surveys"

_forests, doi:10.3390/f10111045_

Round 1

Reviewer 1 Report

The article is about a statistical method that offers expansion factors for small-area based on a probability distribution. The article is interesting and contains a novel approach in my opinion. The structure of the article is organized well and written with an easy to understand language. There are few small corrections that needs to be addressed before the publication:

GREG is not expanded. All abbreviations should be expanded by their first use. USDA in the first paragraph has been never explained. Reference [2] could be replaced with a more related article about continuous mapping. Line 23-29: this classification could be completed. few geospatial interpolation methods needs to be added: 1- polynomial interpolation, 2-probabilistic methods such as logistic regression and Bayesian methods such as probabilistic graphical models, . Figure (1) title: I suggest to add this line (from original text) or something similar to make figure more accurate: Locations are adjusted (not accurate). Line 55, USFS is not expanded. Line 93: “has been more fertile ground for recent innovation than has raking.” : “has” should be removed. 93: this expansion is not allowed, since GREG has been used few times before. Are there many different GREG abbreviations? Line 94: LASSO is not expanded. Line 98: pixel is related to an image. Patch is a better word. Line 104: Suitable reference needs to be added after “…expansion factors.”. Line 116: “Iterative Proportional Fitting” no capital letter is required. Line 118: Generalized Regression -> GREG Lines 174-179 is better to be moved to discussion part. Punctuation mark after equation line 180. Punctuation after equation line 192 Line 252: “the” should be removed. Line 253: Other Agriculture->Agriculture, capitals not necessary.

Author Response

We would like to thank this reviewer for their careful read of the paper and for their comments and suggestions that are certain to improve the paper. The reviewers comments are included below with our reply in bold.

GREG is not expanded. All abbreviations should be expanded by their first use. USDA in the first paragraph has been never explained.

Done.

Reference [2] could be replaced with a more related article about continuous mapping. Line 23-29: this classification could be completed. few geospatial interpolation methods needs to be added: 1- polynomial interpolation, 2-probabilistic methods such as logistic regression and Bayesian methods such as probabilistic graphical models,

We have added the following references to this section.

Healey, S.P., Cohen, W.B., Yang, Z., Brewer, K., Brooks, E.B., Gorelick, N., Hernandez, A., Huang, C., Hughes, M.J., Kennedy, R.E., Loveland, T.R., Moisen, G.G., Schroeder, T.A., Stehman, S.V., Vogelmann, J.E., Woodcock, C.E., Yang, L. and Zhu, Z. (2018). Mapping forest change using stacked generalization: An ensemble approach. Remote Sensing of Environment. 204, 717-728.

Schroeder, T.A., Schleeweis, K.G., Moisen, G.G., Toney, C., Cohen, W.B., Freeman, E.A., Yang, Z., and Huang, C. (2017). Testing a Landsat-based approach for mapping disturbance causality in U.S. forests. Remote Sensing of Environment. 195, 230-243.  **This is already #7 in the citation list

Cushman, S.A., Macdonald, E.A., Landguth, E.L., Malhi, Y., and Macdonald, D.W. (2017). Multiple-scale prediction of forest loss risk across Borneo. Landscape Ecology. DOI 10.1007/s10980-017-0520-0

Kumar, R., Nandy, S., Agarwal, R., and Kushwaha, S.P.S. (2015). Forest cover dynamics analysis and prediction modeling using logistic regression model. Ecological Indicators. 45, 444-455.

Dlamini, W.M. (2011). A data mining approach to predictive vegetation mapping using probabilistic graphical models.Ecological Informatics. 6, 111-124.

. Figure (1) title: I suggest to add this line (from original text) or something similar to make figure more accurate: Locations are adjusted (not accurate). The figure now reads (Plot locations are jittered to protect privacy.)

Line 55, USFS is not expanded.

Expanded at line 15 now.

Line 93: “has been more fertile ground for recent innovation than has raking.” : “has” should be removed.

Done

93: this expansion is not allowed, since GREG has been used few times before. Are there many different GREG abbreviations?

Line 93 is the first instance of GREG in the main text so it is expanded here. We have added an expansion to the abstract also.

Line 94: LASSO is not expanded.Expanded and reference added.

 Line 98: pixel is related to an image. Patch is a better word. We wanted to draw the link to imagery to ground the theory. It now reads “grid cells (such as image pixels)”

Line 104: Suitable reference needs to be added after “…expansion factors.”. Added ref to Bechtold and Patterson

 Line 116: “Iterative Proportional Fitting” no capital letter is required. So Changed

 Line 118: Generalized Regression -> GREG Lines 174-179 is better to be moved to discussion part. The discussion of ancillary data uncertainty is central to our motivation of using regularization methods, so we think that it ought to remain here. We have increased the discussion of uncertainty here, and explicitly mentioned that regularized methods provide one solution.

Punctuation mark after equation line 180. Period added

Punctuation after equation line 192 period added

Line 252: “the” should be removed. So Removed

Line 253: Other Agriculture->Agriculture, capitals not necessary. We have changed the capitals as suggested but retained “other agriculture” as it defines agriculture other than forest.

Reviewer 2 Report

See attached word

Author Response

We would like to thank the reviewer for their careful and thorough reading of this manuscript and for their thoughtful and constructive suggestions. We especially appreciate  their detailed discussion of the paper and it's contributions.

We have copied the reviewers comments below and re responses are in bold.

----------------

The following are notes made during my first reading of the manuscript. I recommend minor additions in the Introduction that address some initial confusion on my part. Later sections resolve that confusion. However, minor additions to the Introduction would aid the student.

Figure 1: Either define terms "FIA Phase II Plot" and "Evaluation Group 2011" or omit them from graphic This figure is changed now.

Line 54: describe "expansion factors" and provide key citation Cited Bechtold and Patterson

Line 55: what is a "power user" Deleted power

Lines 59-60: Provide an example of a "forest area estimate" (e.g. ha of pine forest in South Carolina) and an estimate of "other FIA variable" (e.g., tons of softwood biomass in Southern Coastal Plain) Thank you for this excellent suggestion. It now reads as “Additionally, these expansion factors are designed for estimating the area of forest within a survey unit, and may not be suitable for estimates of other FIA variables, such as ha of pine forest in a survey unit or tons of softwood biomass in Colleton County, South Carolina.

Lines 68-69: Provide example of domain the can be "supported by a direct estimate from plot-level data", such as a "FIA Survey Unit" in Figure 1 \added{Currently, expansion factors only support the direct estimation at the scale of FIA survey units (as in Figure 1).}

Line 81: Give an example of "selecting small enough areas" survey unit? pixel? Rewritten as “A wall-to-wall map can be produced by estimating expansion factors for image pixels or other  small areas.”

Line 88: What is the value of the "map of expansion factors for each plot"? \added{Such a map would describe the probability that any location can be represented by a particular plot.}

Line 97: Clearly state the assumption that you use finite population estimators This sentence not reads ”We seek a finite population estimator for the population and study area $U$ of size $N$, using a sample survey $s$ of size $n$.”

Lines 97-114: It would help me if there was a table that concisely defines the symbols used by the authors Table 1 has been added.

Line 104: Add seminal citation to "expansion factor" Cited Bechtold and Patterson

Line 117: Suggest clarification: “Two widely used methods of accounting for ancillary data are (1) raking, which is also called “Iterative Proportional Fitting” and “poststratification” when there is only one ancillary variable, and (2) Generalized Regression (GREG) The sentence has been rewritten (now at line 126)

Line 120: Does not "plot-level characteristics that we know totals for" equate to auxiliary variables, and if so, why not just call these auxiliary variables? Yes, so changed.

Line 122-125: Complete enumeration of remotely sensed measurements for each and every member of the population appears to be a true census. Those known population totals for wall-to-wall remotely sensed data will differ from the unknown population totals for measurements conducted by field crews. Unless I misunderstand the authors' perspective, I do not agree with their statement "Since most ancillary data have uncertainty, they do not strictly satisfy the condition that ancillary data be known." Those ancillary data only include uncertainty under the assumption that measurements with remote sensing protocols must exactly agree with measurements with the field protocol. It appears that this assumption is unnecessary. The more important relationship is that measurements of auxiliary variables are correlated with measurements of the study variables at the scale of the population unit. That correlation need not be +1.0 or −1.0. However, the stronger the correlation, the stronger the variance reduction with auxiliary data.We think the reviewer has raised an interesting discussion here, and we have tried to incorporate this into the paper. We believe that there are two points brought up by the reviwer: one a general issue about uncertainty in ancillary data, and a second - more technical - point about correlations. We have now tackled the uncertainty question here, and defer the discussion of correlation until the discussion of the paper. On the uncertainty point, we emphatically disagree with the reviewer. In responding to this, we have clarified in what sense GREG and raking require zero uncertainty ancillary data, as without it, the estimators do not achieve consistency in 100 percent samples. Second, we believe that all remotely sensed data do have uncertainty. The only measurement satellites are capable of making are  reflectance at the sensor (top of atmosphere reflectance). Everything else is a modeled output with uncertainty. But we also readily accept the reviewer’s point that field data have uncertainty as well. And as professionals who regularly try to point out to survey methodologists and field workers that survey data are not the “gold standard” they think they are, we are shocked at ourselves for not making this point in the text, and we thank the reviewer for bringing this up. Please see the added discussion here (now at lines 132-140).

Line 126: Perhaps I missed something in the Introduction, but are there multiple expansion factors for each plot? One for each study variable for each plot? One for each plot? A plural was here that shouldn’t have been. But traditional raking has one factor per plot. We create a vector per plot. We now call attention this at line 215

Line 141.1 to 141.2: Are the explanatory variables actually population statistics (totals) for each auxiliary variable?Yes

Line 147: Can negative correlations between study variables and auxiliary variables cause negative expansion factors? Negative expansion factors are impossible in raking owing to the entropy objective function. A comment about this has been added at line 323 that reads “For our purposes, an advantage of the entropy distance function compares to the Chi-square function is that it prevents survey weights from becoming negative.” Negative expansion factors are possible in GREG even with strictly positive correlations.  Mathematically, GREG is a linear regression, not bounded by 0, and sometimes adding negative weights to certain plots is necessary to minimise the sum square error. This is especially so for predicting small counts because a linear model can produce predictions beyond 0.

Line 161: Typographical error ?: Should "if" be "it"? So changed.

Lines 171-173: Up to this point, I had assumed that the regression coefficients forced population estimates for the auxiliary variables (not study variables) to match their known population parameters. Wall-to-wall auxiliary variables provide known population totals, without sampling or measurement error. Does it really matter that they would not be expected to exactly equal the unknown population totals for the study variables? Addressed in earlier comment. We can think of no auxiliary variable (or field measurement, for that matter) that represents any aspect of the earth’s surface without uncertainty.

Line 182: Typographical error: “Guggemos et al. show [25] show that…”extra show deleted

Line 185: What is the role of γparameter in context of regularized raking?Added “The regularization parameter \gamma served to regulate the tradeoff between finding expansion factors that are close to the unbiased design weights and finding expansion factors that reproduce the auxiliary totals.”

Line 192: Please concisely describe how the ’Matrix’ package in R minimizes this function Reference to precise page number for the linear equation in  Guggemos now given.

Line 294: Typographical error: estimate(s) So Changed

Lines 320-321: I do not understand the observation “it is much less reasonable to expect covariances between different ancillary data products” Added “For example, while we might obtain published standard errors for pixel-level estimates of tree canopy cover, we will not get covariances between pixels, nor will we get covariances between tree-canopy estimates and land cover estimates (for example). Of course, many such ancillary predictions will be correlated, but we must disregard aspect here.”

Lines 325-333: It is not apparent to me from this paragraph how cross-validation is used to select the regularization parameter γ.The authors later clarify the methods with the example in lines 358-369. However, a concise preview in lines 325-333 would help the student.This paragraph is rewritten. Now at lines 355 - 363

Line 385: Typographical error-difference(s) So Changed

Line 405: Typographical error-interst So Changed

Lines 410-411 Typographical error? "Because none of the weights are negative, all estimates like strictly within the range of the survey data." So Changed

Line 438: "(E)stimates are relatively design-unbiased"; are estimates or an estimator unbiased? Is not an estimator either biased or unbiased? Do you mean that the degree of bias in the biased estimator is relatively small? Yes, so changed

Line 439: Should “path-level” be “plot-level”?Yes, so changed.